# Structures revealing mechanisms of resistance and collateral sensitivity of *Plasmodium falciparum* to proteasome inhibitors

**Hao-Chi Hsu** [1], **Daqiang Li** [2], **Wenhu Zhan** [2], **Jianxiang Ye** [2], **Yi Jing Liu**[3], **Annie Leung**[3], **Junling Qin**[4], **Benigno Crespo**[5], **Francisco-Javier Gamo** [5], **Hao Zhang**[2], **Liwang Cui** [4,6], **Alison Roth** [7], **Laura A. Kirkman**[2,3], **Huilin Li** [1] ✉ & **Gang Lin** [2] ✉

The proteasome of the malaria parasite *Plasmodium falciparum* (Pf20S) is an advantageous drug target because its inhibition kills *P. falciparum* in multiple stages of its life cycle and synergizes with artemisinins. We recently developed a macrocyclic peptide, TDI-8304, that is highly selective for Pf20S over human proteasomes and is potent in vitro and in vivo against *P. falciparum*. A mutation in the Pf20S β6 subunit, A117D, confers resistance to TDI-8304, yet enhances both enzyme inhibition and anti-parasite activity of a tripeptide vinyl sulfone β2 inhibitor, WLW-vs. Here we present the high-resolution cryo-EM structures of Pf20S with TDI-8304, of human constitutive proteasome with TDI-8304, and of Pf20Sβ6$^{A117D}$ with WLW-vs that give insights into the species selectivity of TDI-8304, resistance to it, and the collateral sensitivity associated with resistance, including that TDI-8304 binds β2 and β5 in wild type Pf20S as well as WLW-vs binds β2 and β5 in Pf20Sβ6$^{A117D}$. We further show that TDI-8304 kills *P. falciparum* as quickly as chloroquine and artemisinin and is active against *P. cynomolgi* at the liver stage. This increases interest in using these structures to facilitate the development of Pf20S inhibitors that target multiple proteasome subunits and limit the emergence of resistance.

Over a third of the world's population is at risk of malaria. Malaria, an infectious disease caused by protozoan *Plasmodium* parasites, has been with humans for thousands of years[1]. Ever since the discovery of quinine, humans have developed many effective antimalarial drugs, yet *Plasmodium* parasites have successfully developed resistance to almost all of them, even to polypharmacological artemisinin (ART) and its close derivatives, the backbone of first-line treatments for non-complicated malaria[2–4]. ART resistance is widespread in the Greater Mekong Subregion[5–7] and has independently emerged and spread in the Sub-Sahara region[8]. Mutations conferring resistance to ART in both

[1]Department of Structural Biology, Van Andel Institute, 333 Bostwick Ave NE, Grand Rapids, MI 49503, USA. [2]Department of Microbiology & Immunology, Weill Cornell Medicine, 1300 York Avenue, New York, NY 10065, USA. [3]Division of Infectious Diseases, Department of Medicine, Weill Cornell Medicine, 1300 York Avenue, New York, NY 10065, USA. [4]Department of Internal Medicine, Morsani College of Medicine, University of South Florida, Tampa, FL 33612, USA. [5]Global Health Medicines R&D, GlaxoSmithKline, Severo Ochoa 2, 28760 Tres Cantos, Madrid, Spain. [6]Center for Global Health and Infectious Diseases Research, College of Public Health, University of South Florida, Tampa, Florida, USA. [7]Department of Drug Discovery, Experimental Therapeutics Branch, The Walter Reed Army Institute of Research, 503 Robert Grant Ave., Silver Spring 20910 MD, USA. ✉e-mail: huilin.li@vai.org; gal2005@med.cornell.edu

regions have been mapped to an allele in the *Kelch13* gene, suggesting a convergent path for the evolution of ART resistance. These clinical observations raise the possibility that the parasites are capable of evolving ART resistance beyond the ring stage, where the current resistance is manifest, and into other stages of their life cycle. Loss of efficacy of ARTs could potentially lead to devastating consequences in public health, such as happened in Africa in the late 1900s when *Plasmodium* parasites developed chloroquine resistance[9].

Proteasomes of pathogens have recently emerged as tractable targets for the development of antimicrobial drugs[10]. Preclinical studies have been launched for proteasome inhibitors targeting proteasomes of *Leishmania* and *Trypanosoma*[11,12]. There are several unique advantages for antimalarial agents in targeting the malaria proteasome, including that: 1) the parasite is hyper-sensitive to the loss of proteasome function; 2) the parasite is susceptible at several major stages of its life cycle, including the erythrocytic, liver, gametocyte, and the male gamete activation stages; 3) proteasome inhibitors have been discovered that can be selective for Pf20S over human constitutive proteasome and human immunoproteasome; and 4) such inhibitors include some that are tractable for pharmacological development[13-17]. At present, however, Pf20S inhibitors that spare both human constitutive (c-20S) and immuno-proteasomes (i-20S) are not orally bioavailable. The only reported orally bioavailable Pf20S inhibitor with efficacy in a humanized mouse model of malarial infection is a peptide boronate that is only modestly selective for Pf20S over c-20S and has no selectivity over i-20S[17]. We recently developed TDI-8304 as a potent, species selective, and noncovalent Pf20S inhibitor that showed efficacy in a humanized mouse model of *P. falciparum* (Pf) infection with subcutaneous dosing[16]. However, it is not orally bioavailable and structure-guided optimization of its pharmacokinetic properties is urgently needed.

Several reports have shown that the minimal inoculum for selection of resistance against proteasome inhibitors ranges from $10^7 - 10^9$, dependent on the modality of the inhibitors[18]. Surprisingly, mutations that confer resistance to β5 inhibitors make the mutants more susceptible to a β2 inhibitor, that is, such strains show collateral sensitivity[18,19]. The reason remains elusive, as the mutated residues are usually distant from the β2 active sites. In this report, we describe structural studies of Pf20S and c-20S with the β5 inhibitor TDI-8304 and of Pf20Sβ6[A117D] with the β2 inhibitor WLW-vs that shed light on the compounds' potency, specificity and selectivity, and the mutant parasite's mechanism of resistance and collateral sensitivity. The availability of these high-resolution structures provides a foundation for structure-based Pf20S inhibitor drug development.

## Results

### TDI-8304 kills *P. falciparum* rapidly, overcomes ART resistance and is active against *P. cynomolgi* in liver stage

We previously reported two classes of non-covalent Pf20S selective inhibitors, one with a fast-on/off kinetics, represented by TDI-4258[19], and one with a time-dependent inhibition kinetics, represented by TDI-8304 (Fig. 1a)[16]. TDI-4258 kills parasites with a modest in vitro parasite reduction ratio similar to that of the antimalarial drug pyrimethamine[19]. TDI-8304 inhibits Pf20S in a time-dependent manner with a dissociation rate constant $k_{off}$ of 0.0008 $S^{-1}$ and a half-life ($t_{1/2}$) of 14.4 min for the enzyme–inhibitor complex[16]. To investigate if inhibitors of the same target with different inhibition mechanisms yield different parasite killing kinetics, we determined the parasite reduction ratio of TDI-8304. TDI-8304 showed fast-killing kinetics comparable to those of fast-acting artesunate and chloroquine (Fig. 1b). This observation suggests that the longer the residence time of an inhibitor on Pf20S, the faster the killing of the parasites. Furthermore, given that the $t_{1/2}$ of the Pf20S–TDI-8304 complex was modest, this result suggests that unimpaired Pf20S function is critical

for parasite cellular functions, such that even transient Pf20S inhibition is highly detrimental to parasite viability.

Mutations in Kelch13 (K13) are associated with ART resistance at the ring stage in laboratory settings[20-22]. In a ring-stage survival assay, the ART-resistant strain Pf Dd2 K13[R539T] is highly resistant to dihydroartemisinin (DHA), whereas the strain Pf Dd2 K13[Rev] with the T539 reverted to R539[23] is highly susceptible to DHA (Fig. 1c). Conversely, the K13[R539T] strain is more sensitive to TDI-8304 than the K13[Rev] strain (Fig. 1c)[23]. Mutants resistant to proteasome inhibitors were found to be more susceptible to ART than the corresponding wild-type strains[19]. Even a strain with double mutations in both the proteasome and K13 was more susceptible to DHA than the wild-type strain[24]. Thus, resistance to ART and DHA on the one hand and to proteasome inhibitors on the other hand appears to be mutually exclusive, making the combination of ARTs and Pf20S inhibitors appealing for antimalarial regimen development.

Additionally, a repetitive exposure of Pf Dd2 to DHA that was replenished every day for 2 years yielded non-Kelch-related ART-resistant clones. These have multiple mutations in different proteins that confer resistance to ART not only at the ring stage but also at the trophozoite stage, which are markedly different from the Kelch13 mutations that confer partial resistance only at the ring stage[6,23]. In a standard parasite growth inhibition assay, these clones are highly resistant to DHA with ~10-fold $EC_{50}$ shift in the erythrocyte assays, but almost no change in $EC_{50}$ values for TDI-8304 (Fig. 1d, e). Thus, a proteasome inhibitor not only overcomes ART resistance based on the most prevalent mutations but would likely also suppress the emergence of ART resistance conferred by other mutations.

To test if TDI-8304 is active against the parasite liver stage, we took advantage of a recently developed *P. cynomolgi* (Pc) liver stage model[25,26]. Because TDI-8304 is rapidly cleared by monkey microsomes, we tested TDI-8304 activity in the liver stage both in the absence and presence of aminobenzotriazole (ABT), which inhibits cytochrome P450's 3A4, 3A4/5, 2E1, 2C19[27]. In the absence of ABT, TDI-8304 showed minimal activity against *P. cynomolgi* hypnozoites and schizonts in prophylactic drug treatment mode and no activity in radical cure mode. However, in the presence of ABT, TDI-8304 showed an $IC_{50}$ of 3.1 μM against hypnozoites and an $IC_{50}$ of 2.4 μM against schizonts in prophylactic treatment mode. Furthermore, TDI-8304 showed an $IC_{50}$ of 1.8 μM against schizonts in simian hepatocytes when treated in radical cure mode (Fig. 1f). The results suggest that TDI-8304 is active against *P. cynomolgi* in the liver stage.

### High-resolution structure of the Pf20S−TDI-8304 complex

To understand the inhibition mechanism and species selectivity of TDI-8304 at a molecular level, we sought to determine the structure of Pf20S with TDI-8304. Structures of Pf20S bound to several inhibitors have been reported at medium resolution range (3.1 Å to 3.6 Å)[15,17]. Docking of TDI-8304 into these structures did not yield insight on the species selectivity. Using highly enriched Pf20S purified from Pf NF54 pellets, we determined the cryo-EM structure of the TDI-8304-bound complex at an average resolution of 2.18 Å (Supplemental Figs. 1–3). The 20S proteasome core particle is composed of two antechambers located between an α-ring and a β-ring and a catalytic chamber centrally located between two β-rings. Unfolded peptide substrates must pass through the antechamber before reaching the catalytic chamber for degradation. In the Pf20S−TDI-8304 structure, in addition to two anticipated TDI-8304 molecules in the two β5 subunits, there are four unexpected TDI-8304 molecules, two in the two β2 subunits, and two in the two antechambers near the β-annulus, in a pocket formed by subunits β3, β4, and α3 (Fig. 2a, b). The EM densities of all six TDI-8304 molecules were well resolved for unambiguous atomic modeling. No TDI-8304 density was observed in any β1 catalytic pocket, even when the EM map was displayed at a low threshold.

**TDI-8304 binds strongly in the β5 site but weakly in the β2 site.** The β5 inhibitor density is in the canonical substrate binding cleft between subunits β5 and β6 (Fig. 2c, d). TDI-8304 binds to the β5 subunit with the P1 cyclopentyl group projecting into the S1 pocket and the P3 morpholino group inserting into the S3 pocket. The main chain of the cyclic peptide-like inhibitor binds to the substrate cleft in an anti-parallel β-sheet mode, forming an extensive hydrogen bonding network with Ser21, Gly47, and Ala49 of β5 and the side chain of Asp153 of β6 (Fig. 2d). Two water molecules connect the TDI-8304 to the β5 Gly23 and the β6 Asp153. The TDI-8304 pyrrolidinone forms two hydrogen bonds with the β6 Ser154 to further enhance the affinity. Importantly, TDI-8304 also makes extensive contact with several hydrophobic residues including Ala20, Met22, Met45, Gly48, and Ala49 of β5 and Phe135, Val155, and Cys159 of β6. These extensive hydrophilic and hydrophobic interactions likely account for TDI-8304's potent inhibition of the Pf20S β5 activity.

In contrast to the potent inhibition of β5 chymotrypsin-like activity, TDI-8304 only affords 30% inhibition of the β2 trypsin-like activity at 100 μM (Supplemental Fig. 4). In our structure, TDI-8304 forms hydrogen bonds with the backbone of the Thr21, Gly47 and Ala49 of β2 and the side chain of β3 Asp138 (Fig. 2e, f). Therefore, the main interaction with β2 is in the antiparallel β-sheet mode that resembles the interaction with β5. In addition, three water molecules coordinate six hydrogen bonds between TDI-8304 and the binding pocket (Fig. 2f). The TDI-8304 P5 pyrrolidinone forms a hydrogen bond with β2 Glu22, leading to a different pose than in the β5 (compare Fig. 2c, e). TDI-8304 binding in the β2 also involves extensive hydrophobic interactions, including Ala20, Val48, and Ala49 of β2 and Leu139, Ile140, and Cys144 of β3 (Fig. 2f).

By superimposing the β2 and β5 substrate binding pockets, we found that the two sites are overall similar, but with notable distinctions in the P2-P4 tether and the P5 pyrrolidinone. (Supplementary

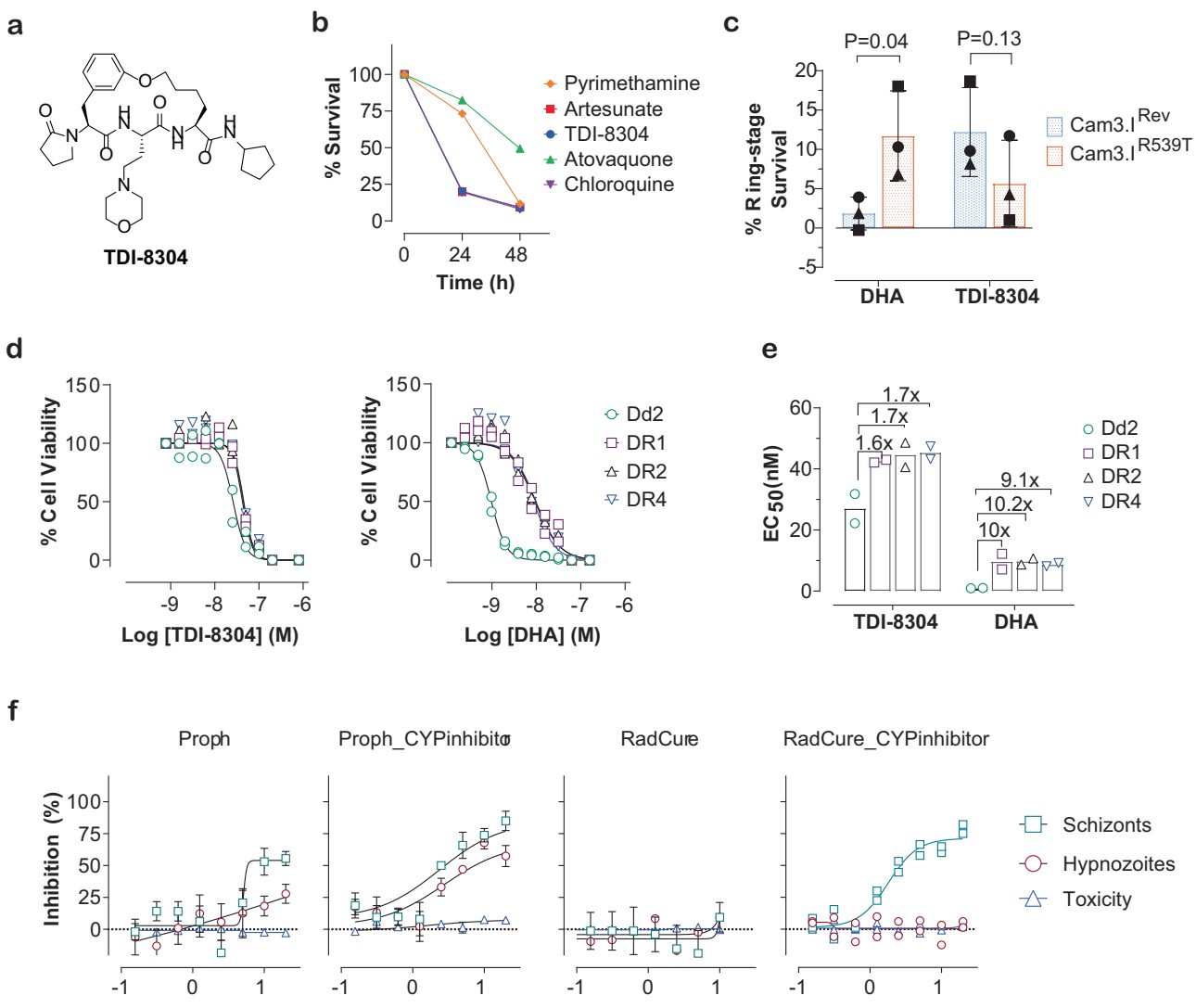

**Fig. 1 | TDI-8304 rapidly kills *P. falciparum* parasites and overcomes ART-resistance at both ring-stage and trophozoite stage. a** Structure of TDI-8304. **b** Parasite reduction ratio of TDI-8304, in comparison with other antimalarial drugs. **c** Cam3.I[R539T] is resistant to ART at ring-stage survival assay but more susceptible to TDI-8304 than Cam3.I[Rev]. Means ± SD from three independent experiments distinguished by shapes of the symbols and compared by paired Student's *t* test. **d**, **e** Dd2 derived ART-resistant DR1/2/4 at trophozoite-stage are highly resistant to DHA but are as sensitive to TDI-8304 as Dd2 (left); dose-dependent growth inhibition of Dd2, DR1/2/4 by TDI-8304 and DHA (right), respectively. Symbols are biological duplicates from two independent experiments distinguished by shapes of the symbols. **f** TDI-8304 is active against *P. cynomolgi* at liver stage in a radical cure model. Dose response plots of inhibition of *P. cynomolgi* liver stage parasites following treatment with TDI-8304; simian hepatocyte nuclei counts normalized to DMSO controls are shown to demonstrate selectivity index. Graph bars represent means with s.d. of experimental and biological replicates (*n* = 2).

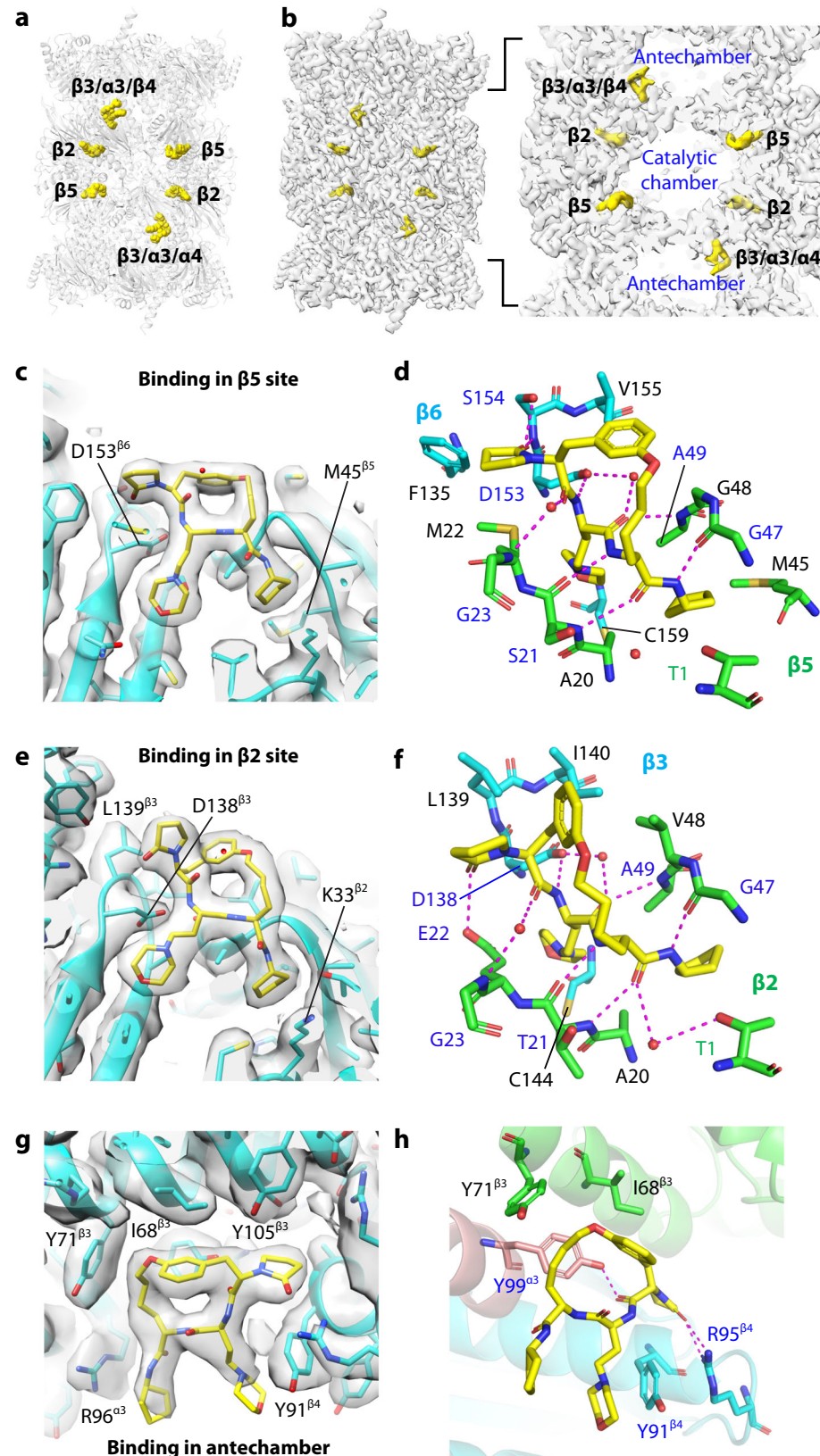

**Fig. 2 | Structure of Pf20S with TDI-8304. a** Overall structure of Pf20S-TDI-8304 with six TDI-8304 molecules shown in yellow spheres. **b** Surface-rendered EM map of the Pf20S-TDI-8304 complex with the inhibitor densities in yellow. Right panel is a zoomed view showing the 4 TDI-8304 molecules in the catalytic β2 and β5 sites, and two molecules in the antechamber in cavities surrounded by β3/α3/β4. **c, d** TDI-8304 binding in the β5 site showing the local EM density (**c**) and the detailed interactions (**d**). The residues for hydrophobic interactions are in black. **e, f** TDI-8304 binding in the β2 site showing the local EM density (**e**) and the detailed interactions (**f**). **g, h** TDI-8304 binding in the antechamber showing the local EM density (**g**) and the detailed interactions (**h**).

Fig. 5a, b). A detailed comparison suggests that the pyrrolidinone and hydrophobic interactions are primarily responsible for TDI-8304's different inhibition activity in these two sites. In the S2 and S4 sites, the side chain of β2 hydrophobic residue Val48 is bulkier than β5 G48 and seemingly pushes the P2-P4 tether towards the center of the pocket in the β2 site. The P5 pyrrolidinone has only a weak hydrogen bonding with β2 Glu22, as compared to the strong hydrophobic interactions with β5 Met22 and Phe135 and two hydrogen bonds with β6 Ser154 in the β5/β6 site. In the S3 site, β2 Cys144 is 1.6 Å further away compared to the β5 Cys159 and therefore interacts with the P3 morpholino group more weakly than the β5 Cys159. In the S1 site, the P1 group is in the same conformation, but the β5 Met45 hydrophobically interacts with the P1 cyclopentyl moiety, which is lacking in β2 with Gly45 at this site. These structural details explain the much weaker inhibition of β2 by TDI-8304 than its inhibition of β5.

**TDI-8304 binding in the antechamber.** We were surprised to observe TDI-8304 in the antechamber near the β-annulus (Fig. 2a, b). We are unaware of another example in which a 20S inhibitor also occupies a site in the antechamber. The macrocyclic ring of TDI-8304 fits snugly in an alcove formed by the α3, β3, and β4 subunits, with the P1 cyclopentyl and P3 morpholino groups partially exposed to the solvent in the antechamber (Fig. 2g, h). There are three hydrogen bonds, one is between a TDI-8304 main chain oxygen and α3 Tyr99, and the other two between the pyrrolidinone and β4 Arg95. TDI-8304 also forms hydrophobic interactions with Ile68 and Tyr71 of β3 and Tyr91 of β4. It is unclear if the antechamber binding of TDI-8304 is fortuitous or contributes to the potent anti-parasite activity. Since a protein substrate must pass through the antechamber before reaching the catalytic chamber, an inhibitor's binding in the antechamber may influence the substrate passage. However, the antechamber binding site is at the side and away from the direct substrate approach.

Interestingly, the outer surface of the Pf20S antechamber is unique among the eukaryotic 20S core particles in having extensive asparagine repeats in the β1 and β7 subunits (Supplementary Fig. 6a). Asparagine-rich, low-complexity regions are often found in many *P. falciparum* proteins[28]. We found that the four-residue (Leu70 to Pro73) short loop in human β1 is replaced by a 57-residue (Asn70 to Asn126) long loop in the Pf20S β1 (Supplementary Fig. 6b). The 6-residue loop in the human β7 is replaced by a 42-amino-acids long loop in Pf20S β7 (Supplementary Fig. 6c). These extended loops are rich in asparagine and acidic amino acids, have well defined EM densities, and can be unambiguously modeled. The first long loop is positioned among the β1, α1, and α7 subunits, the second loop is among the β7, α6, and α7 subunits. Together they appear to strengthen the interface between the β-ring and the α-ring, perhaps enhancing the structural integrity of the Pf20S core particle.

**Structural determinant for species selectivity of TDI-8304**
Species specificity is critical in antimicrobial drug development. To understand TDI-8304's specificity for the Pf20S over the human proteasomes, we compared the Pf20S-TDI-8304 structure with the c-20S complexed with Bortezomib at 2.10 Å[29] and the i-20S complexed with M3258 at 2.29 Å[30] (Fig. 3a, b). We found that there is no steric conflict that prevents TDI-8304 from binding in the β2 catalytic pockets of both c-20S and i-20S (Fig. 3a). In computationally docked poses, TDI-8304 forms six and four H-bonds with c-20S and i-20S β2 subunits, respectively, comparable to the six non-water mediated H-bonds with Pf20S β2 in the experimental structure. Because TDI-8304 only weakly binds Pf20S β2, we expect the inhibitor also binds weakly to the c-20S or i-20S β2 subunits, suggesting that TDI-8304's species selectivity is not determined by its binding in the β2 pocket, and therefore, is likely determined by structural features in the β5 binding pocket. Furthermore, the P5 pyrrolidinone of TDI-8304 is only 2.3 Å away from the β6 Pro126 in both c-20S and i-20S β5 substrate pockets, and the phenyl

ring of the P2-P4 tether is only 2.2 Å away from the β5 Cys48 of the i-20S (Fig. 3b). These potential steric conflicts may lower the inhibitor binding affinity in the β5 catalytic pockets of c-20S and i-20S, explaining the lack of inhibition of the human proteasomes.

To experimentally determine the selective mechanism against the human proteasome, we solved a cryo-EM structure of human c-20S in complex with TDI-8304 (c-20S-TDI-8304). The cryo-EM map was refined to an average resolution of 2.0 Å (Supplemental Figs. 9, 10). The atomic model shows that TDI-8304 is in the β5 pockets (Fig. 3c), and no extra density was observed in the β2 pockets nor the antechambers. In the β5 pocket, TDI-8304 binds in an antiparallel binding mode (Fig. 3d), which is consistent with the computationally docked model. However, the steric hindrance from Pro126 of c-20S β6 forces the P5 pyrrolidinone group of TDI-8304 to flip downwards and toward the cyclic ring, resulting in a loss of two hydrogen bonds from pyrrolidinone compared with TDI-8304 in the β5 of the Pf20S wild type. Based on our analysis of the TDI-8304 binding in the Pf20S β5 site with high affinity and in the Pf20S β2 and c-20S β5 sites with low affinity, we suggest that the two hydrogen bonds between pyrrolidinone of TDI-8304 and Ser154 of Pf20S β6 play a pivotal role in driving the high-affinity binding of TDI-8304 to the Pf20S β5 pocket.

**Structure of the Pf20Sβ6^A117D–WLW-vs complex**
We previously showed that the A117D mutation in the Pf20S β6 subunit caused an 18-fold increase in EC_{50} of the TDI-8304 against Dd2, whereas the A49S mutation in the catalytic β5 pocket caused no change in the parasite's susceptibility to the compound[16]. Further, the β6A117D mutation enhanced the activity of the β2-specific inhibitor WLW-vs against β2 enzymatic activity and anti-parasite activity[19]. Because β6A117 is ~55 Å away from the β2 on the same β ring and ~40 Å away from the β2 catalytic site on the different β ring, it has been a mystery how the β6 mutation causes opposite effects at β5 and β2 sites. To answer this question, we determined a cryo-EM structure of the partially purified β6A117D mutant Pf20S (Pf20Sβ6^A117D) with WLW-vs at 2.58 Å resolution (Supplemental Figs. 7, 8). Because WLW-vs is a β2-specific inhibitor, we were surprised to observe WLW-vs in both β2 and β5 catalytic sites of the Pf20Sβ6^A117D (Fig. 4a, b). The overall binding mode of WLW-vs in both sites is similar, following an anti-parallel β-strand configuration resembling peptide substrates (Fig. 4c, d). In both β2 and β5, the inhibitor is covalently linked to the catalytic Thr-1, the two indole groups at the P1 and P3 sites are inserted into the S1 and S3 pockets, respectively, and the P2 Leu and P4 morpholine face the catalytic chamber. Superimposition of the β2 and β5 pockets reveals that WLW-vs binds differently in the two catalytic subunits (Fig. 4e).

We previously hypothesized that the β6A117D mutation likely causes conformational changes around A117 due to the tightly spaced around the A117[19]. Tyr residues 150 and 158 in β6 could undergo conformational changes with the Ala-to-an-acidic-Asp mutation. We compared the β6 subunit structures of the wild-type Pf20S and A117D mutant Pf20S (Fig. 5a–c), and with the high resolution of the two EM maps, we clearly see that the β6A117D caused a one-residue shift as well as an upside-down flip in the Gly156-to-Ala161 region. In the wild-type β6 structure, Tyr158 points upward to interact with the small hydrophobic Ala117. In the A117D mutant β6 structure, the negatively charged Asp117 has expelled Tyr158 from its wild-type position, causing the observed one-residue shift and the upside-down flip in the Gly156-to-Ala161 region (Fig. 5c). This conformational change has two profound consequences: collateral sensitivity to the WLW-vs β2 inhibitor and resistance to β5 inhibitors.

**Resistance.** β6-Tyr158 in the Pf20Sβ6^A117D mutant structure was found to exist in two alternative conformations (Fig. 5b, c). Both conformations limit TDI-8304 access to the β5 binding pocket. Specifically, the Cβ atom of Tyr158 is only 2.6 Å away from the P3-morpholino, and the

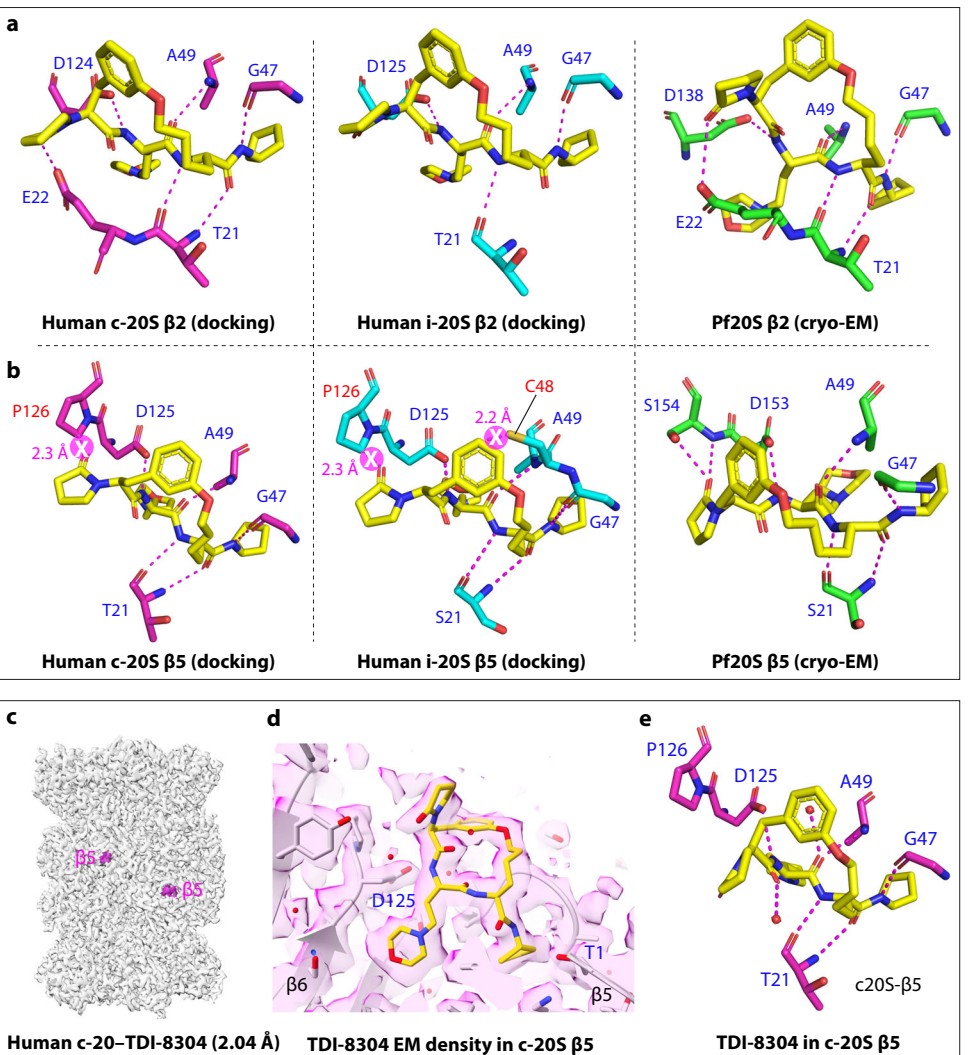

**Fig. 3 | Alignment of Pf20S and human 20S proteasomes. a** The β2 binding pocket of Pf20S-TDI-8304 was aligned with those of human c-20S (PDB ID 5LF3) and i-20S (PDB ID 7AWE) and shown separately. **b** The β5 binding pocket of Pf20S-TDI-8304 was aligned with those of c-20S and i-20S and shown separately. The clash sites are indicated magenta circles with white crosses. The β5 Pro126 of c-20S clashes with the pyrrolidinone of TDI-8304, and the β5 Pro126 and Cys48 of i-20S clash with pyrrolidinone and P2-P4 tether of TDI-8304, respectively. In panels **a** and **b**, the water-mediated H-bonds between TDI-8304 and Pf20S were not shown, to provide a fair comparison with the computationally docked TDI-8304 binding poses in the human 20S particles. **c** EM map of the c-20S complexed with TDI-8304 at 2.04 Å overall resolution. The inhibitor densities (magenta) were found only in the β5, but not in any other sites. **d** EM density of the TDI-8304 in a β5 site. **e** Interactions of TDI-8304 with the human β5 site.

side chain hydroxyphenyl group of Tyr158 in one conformation protrudes into the S3 binding pocket and sterically clashes with the P3-morpholino moiety of TDI-8304 (Fig. 5d). Furthermore, the flipping of the Gly156-to-Ala161 peptide moves Ser154 away from pyrrolidinone and diminishes the TDI-8304's affinity for the β5 pocket (Fig. 5e). These observations likely explain why the β6 A117D mutant parasite is resistant to the β5 inhibitor TDI-8304.

**Collateral sensitivity.** In the β2 catalytic pocket, we identified seven H-bonds between the WLW-vs peptide backbone and the surrounding Thr21, Glu22, Gly47, Ala49 of β2 subunit and Asp138 of β3 subunit. Importantly, there is a water molecule near the β3 Asp138 that forms two H-bonds with WLW-vs, and the inhibitor's sulfonyl group forms two H-bonds with Gly47 and Ser129 to further stabilize the compound in the β2 pocket (Fig. 5f). Additionally, the morpholine ring is held by the hydrophobic interactions with Val48 of β2 and Leu139 and Ile140 of β3, and the indole ring in the S3 site is stabilized by hydrophobic interactions with Ala27 of β2 and Ala142 and Cys144 of β3 (Fig. 5f). In the β5 pocket, the WLW-vs backbone forms five H-bonds with Ser21,

Gly47, Ala49 of β5 and Asp153 of β6, and the sulfonyl group also forms H-bonds with the β5 Gly47 and Ser130, and the WLW-vs morpholine shifts away from its position in β2 to form hydrophobic interaction with Phe135 of β6 (Fig. 5g). Also, the aromatic ring of the Tyr158 at its new position hydrophobically interacts with one indole ring of the WLW-vs. Especially, the hydroxyphenyl ring of Tyr158 interacts with the indole ring of the P3 Trp of the WLW-vs via an offset π-π stacking with a favored vertical distance 3.4 Å (Fig. 5h), which likely explains why the β2-specific inhibitor WLW-vs also binds in the β5 pocket of the Pf20Sβ6[A117D], supporting the observed enhanced inhibition against the β5 activity of the Pf20Sβ6[A117D][19].

## Discussion

There are three FDA-approved anticancer drugs targeting the proteasome, and several proteasome inhibitors for treating Leishmaniasis and Chagas disease are in pre-clinical studies[11,12,31]. The essentiality of Pf20S and the synergy between Pf20S inhibitors and the ART drugs make Pf20S an ideal target for antimalarial drugs, because the risk of the parasites developing resistance to Pf20S inhibitors can be

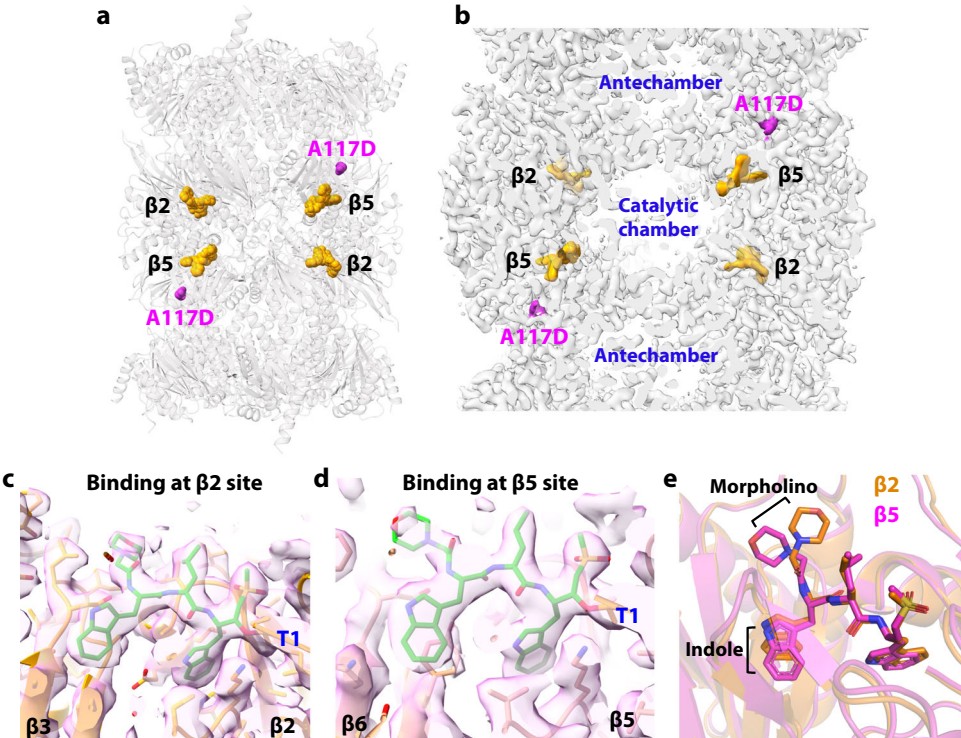

**Fig. 4 | Structure of Pf20Sβ6^{A117D} with WLW-vs. a** Overall structure of the Pf20Sβ6^{A117D}–WLW-vs complex. The four bound inhibitor molecules are shown in orange spheres, and the β6 A117D mutation sites are highlighted in magenta spheres. **b** A cut-open view of the EM map of the complex showing the locations of the inhibitors and the β6 A117D mutation. **c, d** The EM densities of WLW-vs inside the β2 (**c**) and β5 (**d**) catalytic pockets. **e** Alignment of β2 and β5 catalytic pockets shows the altered positions of the inhibitor's morpholino and indole groups at the two sites.

mitigated when used together with the ART drugs. Several studies have found that proteasome mutants that are resistant to proteasome inhibitors are more susceptible to ART treatment and vice versa: for example, Pf K13^{R539T} is more sensitive to proteasome inhibitors than wild type[24]. Thus, Pf20S inhibitors hold the potential to reduce the emergence of resistance to and spread of ART resistance. Many efforts have been poured into the development of oral Pf20S-selective inhibitors, with some success. Peptide boronate MPI-13 is optimized to be orally bioavailable but is only modestly selective in targeting Pf20S over human constitutive proteasome and non-selective in being equally potent against human immunoproteasome[17]. TDI-8304 is highly selective over both constitutive and immuno-proteasomes and efficacious in a humanized mouse model of *P. falciparum*[16], but not orally available. Neither is a drug candidate in its present form; further optimization is needed.

In this study, we showed that a highly selective, noncovalent, slow-binding inhibitor exhibits a fast parasite-killing rate, encouraging its further development. The high-resolution Pf20S structures offer useful insights toward this end. In addition to potent Pf20S β5 inhibition, TDI-8304 also binds to the β2 active subunit. Although TDI-8304 only inhibits β2 activity weakly, its binding pose in β2 is similar to that in β5. We have previously shown that co-inhibition of the β5 and β2 is synergistic[19]. It is possible that even a weak inhibition of the β2 would significantly enhance antimalarial activity resulting from inhibition of the β5. The binding poses of the TDI-8304 in β5 and β2 resolved in this study may provide clues for designing inhibitors that could target both subunits with comparable potency, thus maximizing the potential of the synergy from β2β5 co-inhibition. It is also possible that such co-inhibition could reduce the frequency of resistance.

The structure of the Pf20Sβ6^{A117D} with the β2 inhibitor WLW-vs revealed the mechanism of resistance of the mutant against the β5 inhibitor. The 150° flip of the stretch of 6-amino acids results in Tyr158 being part of the S3 pocket of β5, whilst Ser157 now forms hydrogen

bonds with Asp117. In the β5 active sites of the superimposed model of the Pf20Sβ6^{A117D} with TDI-8304, lacking pyrrolidinone interactions and the clashes between the aromatic ring of Tyr158 and the P3-morpholino moiety of TDI-8304 likely contribute to the reduced enzyme inhibition and anti-parasite activity of the compound. However, this mutation could be less detrimental to proteasome inhibitors with less bulky P3 or amino acid boronate proteasome inhibitors without P3[18]. By comparing the structures of the Pf20S WT and Pf20S A117D, we unambiguously found that the A117D mutation not only causes a conformational change in the β5 inhibitor binding pockets, but also a substantial conformational change remotely in the β2 active subunit that enhances the binding affinity of a β2 inhibitor. However, it is not clear whether the collateral sensitivity would pertain to other β2 inhibitors lacking an aromatic ring that binds into the S3 pocket.

Resistance to a malarial proteasome inhibitor is likely to emerge if the drug used alone. It may be possible to reduce the frequency of resistance by combining a proteasome inhibitor with synergistic artemisinins, possibly through a hybrid of an artemisinin and a proteasome inhibitor[32]. This strategy would not only mitigate resistance to new proteasome inhibitors but also to the artemisinin derivatives currently in clinical use.

## Methods
### Ethics statement
The USAMD-AFRIMS Institutional Animal Care and Use Committee and the Animal Use Review Division, U.S. Army Medical Research and Materiel Command, reviewed and approved this study (PN 22-10). Animals were maintained in accordance with established principles under the Guide for the Care and Use of Laboratory Animals eighth edition[33] and the Animals for Scientific Purposes Act (National Research Council of Thailand. 2015. Animals for Scientific Purposes Act. Government Gazette, Bangkok, Thailand) and its subsequent regulations. The USAMD-AFRIMS animal care and use program is fully

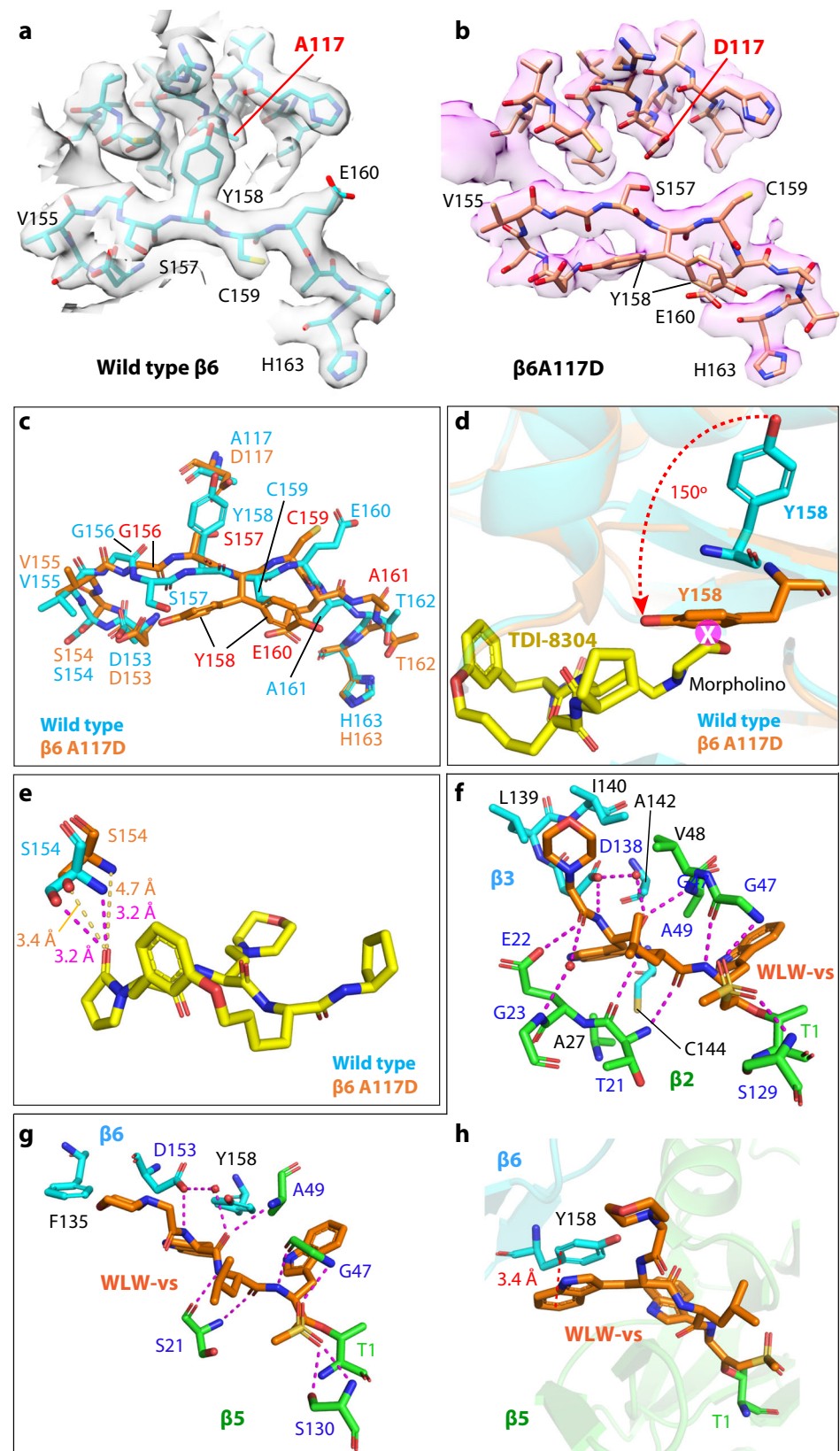

accredited by the Association for Assessment and Accreditation for Laboratory Animal Care International (AAALACi).

## Antimalarial activity in erythrocytic stage

Parasite growth inhibition assays for *P. falciparum* wild type and mutants were performed as reported[19,32].

## Ring-stage survival assay

Ring-stage survival assays (RSA) were performed as reported[32]. Parasite cultures, artemisinin resistant Cam3.I[R539T] and the genetically engineered artemisinin sensitive revertant Cam3.I[rev], were synchronized with 5% sorbitol and a Percoll-sorbitol gradient to obtain tightly synchronized late-stage parasites. After isolation, the tightly

**Fig. 5 | Conformational changes induced by the β6A117D mutation. a, b** The EM densities of the β6 subunit around the Ala117 region in the WT (**a**) and in the A117D mutant Pf20S structure (**b**). **c** Conformational changes induced by the β6A117D mutation. The 6-residue peptide G156-S157-Y158-C159-E160-A161 flips upside down to prevent Tyr158 from clashing with the charged and bulky Asp117. **d** The A117D mutation forces Tyr158 to swing 150° and insert into the S3 site of the β5 catalytic pocket, where the residue clashes with the P3 morpholino group of TDI-8304. **e** Ser154 moves away and loses interaction with the TDI-8304 pyrrolidinone in the A117D mutant β6. The clash with the morpholino group and the loss of interaction

with the pyrrolidinone likely explain the reduced β5 inhibition of the mutant Pf20S, and therefore, the parasite's resistance to TDI-8304. **f** Detailed interactions of WLW-vs in the β2 catalytic pocket. Several hydrophobic residues surrounding the morpholino and the indole groups are in black. **g** Detailed interactions of WLW-vs in the β5 catalytic pocket. In the mutant structure, the β6 Tyr158 flips over to interact with the inhibitor indole group at the S3 site and create a hydrophobic environment. **h** Offset π-π stacking between the indole ring of WLW-vs and the hydroxyphenyl group of Tyr158.

---

synchronized parasites were then allowed to reinvade fresh red blood cells for 3 h and the microscopically confirmed ring stage parasite cultures were again subjected to 5% sorbitol to obtain 0–3 h rings. The isolated ring stage cultures were then plated into a 96-well plate at 0.5% parasitemia and treated with DHA and TDI-8304 at 700 nM at 37 °C in standard gas conditions for 6 h. The plates were then spun and washed to remove compounds and replenished with fresh medium without compounds. Parasite growth was assessed 66 h later using flow cytometry (Becton-Dickson flow cytometer) and nucleic acid stains HO (Hoechst 33342) and TO (thiazole orange).

### *P. cynomolgi* in vitro liver stage assay

Cryopreserved primary simian hepatocytes (donor lots WEG and WXY) and hepatocyte culture medium (HCM) (InVitroGro CP medium) were obtained from BioIVT, Inc. (Baltimore, MD, USA) and thawed following manufacturer recommendations. Hepatocytes were seeded in 384-well collagen-coated plates as previously described[25,26]. Infectious sporozoites were obtained from *Anopheles dirus* mosquitoes infected with *P. cynomolgi* bastianellii (B) strain and were used to infect the plated primary simian hepatocytes 2 days post seed (dpi)[25,26]. TDI-8304 compound was dissolved in dimethyl sulfoxide (DMSO) and used in an 8-point, 2-fold serial dilution at final 10 μM to μM concentrations. The compound was administered in prophylactic or radical cure treatment modes where drug (with or without ABT exposure) was present starting 1 dpi or 4 dpi and repeated daily over a 4-day drug course[25,26]. On 8 dpi, hepatocyte cultures undergo fixation and immuno-fluorescent staining as previously described[25,26,34–36]. Imaging and image analysis of assay plates were completed using an Operetta CLS high-content imaging system and Harmony 4.9 software (Perkin Elmer, Waltham, MA, USA). Using similar methodology described previously, liver stage parasites were quantified by signal emitted by a *Plasmodium*-specific Glyceraldehyde-3-Phosphate Dehydrogenase (GAPDH) antibody and were distinguished as hypnozoite or schizont by area, mean intensity, maximum intensity, and cell roundness[25,26]. Hepatocyte nuclei were quantified by object detection measuring Hoechst stain intensity and a declumping algorithm built-in the Harmony 4.9 software. Calculations for percent inhibition of parasite populations and toxicity of hepatocytes were normalized to controls and dose–response curves were fitted to generate inhibitory concentration (IC$_{50}$) in GraphPad Prism (GraphPad software, Inc., US, Windows version 9.5.1) using the average of experimental and biological replicates.

### Purification of Pf20S

Parasite pellets were thawed on ice with the addition of 2x volume of lysis buffer (25 mM Tris-HCl, 5 mM MgCl$_2$, 1 mM DTT, pH 7.4) for 1 hour, and vortexed vigorously every 5 minutes. The thawed cell mixtures were lysed by sonication with 2 rounds of 10 secs burst (20% power) and 1 min rest on ice. Cell lysate was cleared by centrifugation at 18,407 RCF(g) RPM for 20 min. The resulting supernatant was concentrated using a concentrator with a molecular weight cutoff of 100-kDa, and the buffer was exchanged to Buffer A (20 mM Tris-HCl, 1 mM DTT, pH 7.4). The concentrated sample was run on a tandem 2x HiTrap DEAE Sepharose FF (5 mL each) and fractions containing the LLVY-AMC hydrolyzing activity were pooled, concentrated, buffer exchanged to 25 mM Tris-HCl, pH 7.4, 150 mM NaCl, 1 mM DTT. The resulting sample

was then run on a Superose 6 increase 10/300 GL column. Fractions with the LLVY-AMC hydrolyzing activity were pooled and concentrated for cryo-EM studies.

### Parasite reduction rate (PRR) analysis

Rate of killing was assessed using a standardized method[37]. Briefly, unlabeled erythrocytes infected with the 3D7A (BEI resources) strain were incubated in the presence of the test compound at a concentration corresponding to 10× the EC$_{50}$ determined previously using the 48 h 3H-hypoxanthine incorporation assay (EC$_{50}$ 28 nM; rate of killing determined at 280 nM). Parasites were drug-treated for 24 and 48 h. Tested compounds were renewed after the first 24 h of treatment by removing old media and replenishing with new culture media with fresh drug. After treatment, the compound was washed out and the culture was diluted (1/3rd dilution) using fresh erythrocytes (2% hematocrit) previously labeled with CFDA-SE (carboxylfluorescein diacetate succinimidyl ester, Life Technologies). CFDA-SE labeled erythrocytes were prepared by incubating 1% hematocrit in RPMI 1640 media with 10 μM CFDA-SE at 37 °C for 30 min then washing the cells twice with media and maintaining at 50% hematocrit at 4 °C for up to 24 h before use. Following a further 48 h incubation in standard conditions, the ability of treated parasites to establish new infections in fresh labeled erythrocytes was detected by quantification of double-stained erythrocytes using two-color flow cytometry (Attune NxT Flow Cytometer, ThermoFisher) after labeling of parasite DNA with Hoechst 33342 (Sigma). The Hoechst 33342 is excited by a laser at 405 nm and detected by a 440/50 filter (VL1). CFDA-SE is excited by a blue laser at 488 nm and detected by a 530/30 filter. Samples were analyzed using the Attune NxT software package. Parasite viability is shown as the percentage of infected CFDA-SE stained erythrocytes in drug-treated samples at 24 or 48 h using labeled erythrocytes and labeling of parasite DNA from untreated cultures as a control. Chloroquine, pyrimethamine, atovaquone, and artesunate were used in each assay to validate the assay and allow for a comparative classification of the killing rate of the tested compound. The assay was performed twice, with biological triplicates each time. Experiments were carried out with strain 3D7A (BEI Resources, MRA-151, contributed by David Walliker). The human biological samples were sourced ethically, and their research use was in accordance with the terms of the informed consent under an IRB/EC-approved protocol.

### Cryo-EM sample preparation and data collection

Cryo-EM studies were performed at the cryo-EM core facility of Van Andel Institute. Human c-20S was purchased from R&D system (catalog no. E-360). TDI-8304 dissolved in dimethyl formamide (DMF) was added to Pf20S (0.5 mg/ml) or c-20S (4 mg/ml) in 20 mM Tris, pH 7.5, 5 mM MgCl$_2$, 100 mM KCl, and 1 mM DTT to a final concentration of 0.8 mM of TDI-8304 and 2% of DMF, respectively. To assemble the Pf20Sβ6$^{A117D}$–WLW-vs complex, WLW-vs dissolved in DMF was added to partially purified Pf20Sβ6$^{A117D}$ to a final concentration of 1 mM of the inhibitor and 1% of the DMF. We found that the organic solvent at 1–2% concentration did not significantly affect the particle image contrast. The mixtures were incubated at 37 °C for 60 min before grid preparations. We used Vitrobot Mark IV (FEI, USA) for preparing cryo-EM grids and set the blotting chamber temperature to 6 °C and the relative

humidity to 100%. 3-μl droplets of mixtures of the proteasome–inhibitor complexes were applied to Quantifoil R 1.2/1.3 300 mesh Au holey carbon grids that had been freshly glow-discharged with $O_2$-air for 30 s at 30 W in a Gatan Model 950 Advanced Plasma System. After incubation for 5 s, the grids were blotted with a piece of 595 filter paper with blotting force set to 3 (arbitrary unit) and blotting time set to 3 s. The blotted grids were immediately vitrified by plunging into liquid ethane precooled with liquid nitrogen. Cryo-EM micrographs were recorded in a Titan Krios (Thermo-Fisher, USA) equipped with a Gatan BioQuantum 967 energy Filter and a post-GIF Gatan K3 summit direct electron detector. Movies were recorded using SerialEM software in super-resolution counting mode with a defocus range of -1.0 to -1.4 μm for the c-20S-TDI-8304 complex and -1.3 to -1.8 μm for the Pf20S–TDI-8304 and Pf20Sβ6[A117D]–WLW-vs complexes and a pixel size of 0.414 Å/pixel. The dose rate for the Pf20S–TDI-8304 dataset was 44 electrons/$Å^2$/s. With a total exposure of 1.5 s, the accumulated total electron dose was 66 electrons/$Å^2$. For the Pf20Sβ6[A117D]–WLW-vs dataset, the dose rate was 46 electrons/$Å^2$/s, and with a 1.3 s total exposure time, the final dose was 60 electrons/$Å^2$. For the c-20S-TDI-8304 dataset, the dose rate was 58 electrons/$Å^2$/s. With a 1.0 s total exposure time, the final dose was 58 electrons/$Å^2$. A total of 29,343, 24,806, 16,114 movie micrographs were collected for Pf20S-TDI-8304, Pf20Sβ6[A117D]–WLW-vs, and c-20S-TDI-8304 complexes, respectively.

### Cryo-EM data processing

For Pf20S–TDI-8304, we processed the Pf20S–TDI-8304 dataset with Relion 3.0 and CryoSPARC 2.8[38,39]. We first corrected the image drift with MotionCor2[40] by binning the movie micrographs by a factor of 2 in Relion. The corrected micrographs were exported to CryoSPARC for contrast transfer function (CTF) estimation and correction using CTFFIND4[40]. A combination of manual picking, template picking, and Topaz picking[41] was used for extracting 657,066 particle images from a total of 29,262 raw micrographs. We performed reference-free 2D classification and selected 331,398 particles belonging to 2D class averages with good structural features for ab initio 3D reconstruction and classification. One 3D class with 320,837 proteasome particles had good structural features, and this class was subjected to further classification by 3D multi-reference heterogeneous refinement with 3 classes. Two resulting classes with a total of 305,581 particles had the best structural features, and they were combined for next round of 3D homogeneous refinement, leading to a 3D map at 2.78 Å resolution (C1, no symmetry) and a 2.68 Å 3D map by applying the expected C2 symmetry. We subsequently performed CTF refinement and Bayesian polishing in Relion, followed by masking and automatic B-factor sharpening. The resolution of the 2-fold symmetric 3D map was improved to 2.34 Å after CTF refinement and to 2.18 Å after Bayesian polishing. The resolutions were estimated by applying a soft mask on the EM map and were based on the 0.143 threshold of the gold standard Fourier shell correlation (FSC) of two independently constructed 3D half-maps.

We processed the Pf20Sβ6[A117D]–WLW-vs dataset in Relion 3.1.1 and CryoSPARC 3.3.2[38,39]. The image drift correction, CTF estimation and correction, and particle picking were done similarly to the above Pf20S–TDI-8304 dataset. A total of 23,696 micrographs were used for particle picking. Blob picking was used to generate templates for automated template-based particle picking, which was followed by Topaz training and particle picking of 337,322 particles. 2D classification and ab initio 3D reconstruction led to a selection of 87,879 particles for 3D multi-reference heterogenous refinement. 74,883 particles in the high-resolution 3D class were selected for homogeneous refinement, followed by another round of non-uniform refinement[39]. The final 3D EM map of the Pf20Sβ6[A117D]–WLW-vs complex had an average resolution of 2.58 Å after applying C2 symmetry. The resolution was estimated by applying a soft mask on the map and was based on the gold standard Fourier shell correlation (FSC) at the threshold of 0.143.

The c-20S-TDI-8304 dataset was processed with Relion 4.0.1 followed by CryoSPARC 4.2.1. The image drifts of the movie micrographs were corrected with MotionCor2[40] by a binning factor of 2 in Relion. The corrected micrographs were exported to CryoSPARC for CTF estimation and correction using Patch CTF. The procedures of particle picking were the same as those used in Pf20Sβ6[A117D]–WLW-vs dataset. A total of 2,072,882 particles were picked for reference-free 2D classification. After 2D classification, 1,579,333 particles with good proteasome features were selected for multi-reference heterogenous refinement. 15,310 particles from 2D classification were used to generate a reference for heterogenous refinement using Pf20S as the template. After two rounds of heterogenous refinement, 1,250,898 particles were selected for homogeneous refinement followed by non-uniform refinement. The final resolution was estimated to be 2.04 Å for the 3D EM map reconstructed in the C2 symmetry.

### Model building

We used the published Pf20S atomic model (PDB ID 6MUW) as an initial model to build the atomic structure in the EM map of the Pf20S–TDI-8304 complex[42]. For modeling the TDI-8304 structure, we generated the restraints file using the eLBOW program in Phenix[43]. We then used the atomic model of our Pf20S–TDI-8304 as an initial model to build the atomic model in the EM map of the Pf20Sβ6[A117D]–WLW-vs complex. The starting atomic model of WLW-vs was extracted from the published structure (PDB ID 5FMG). For the c-20S-TDI-8304 structure, we used the published structure (PDB ID 5LF3) as the initial model[29]. All amino acids and inhibitors were manually fitted into the EM maps in Coot[44] followed by real space refinement in Phenix[43,44]. The model quality was estimated using MolProbity in Phenix[43].

### Reporting summary

Further information on research design is available in the Nature Portfolio Reporting Summary linked to this article.

## Data availability

All density maps and models have been deposited in the EMDB and the PDB. The PDB (EMD) IDs are 8G6E (EMD-29764) for Pf20S-TDI-8304, 8UD9 (EMD-42148) for c-20S-TDI-8304, and 8G6F (EMD-29765) for Pf20Sβ6[A117D]-WLW-vs. Source data are provided with this paper.

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

## Acknowledgements

Cryo-EM datasets were collected at the David Van Andel Advanced Cryo-Electron Microscopy Suite at Van Andel Institute. We thank G. Zhao and X. Meng for assisting the data collection. We thank Dr. Carl Nathan for insightful discussions. This research was supported by NIH Grant AI143714 (to G.L.) and AI070285 (to H.L.). AR acknowledges funding in part from the Congressionally Directed Medical Research Programs (W81XWH-WH2210520; PIs D. Fidock and M. Bogyo). The following reagents were obtained through BEI Resources, NIAID, NIH: *Plasmodium falciparum*, Strain Dd2_R539T, MRA-1255, Strain Cam3.I_rev, MRA-1252, contributed by David A. Fidock. The material presented here has been reviewed by the Walter Reed Army Institute of Research. There is no objection to its presentation and/or publication. The opinions or assertions contained herein are the private views of the authors, and are not to be construed as official, or as reflecting true views of the Department of the Army or the Department of Defense or its components.

## Author contributions

H.C.H. performed the cryo-EM data and undertook reconstruction analyses and refined structures; D.L., W.Z., J.W. Y.J.L., A.L., J.Q., B.C., F-J. G., H.Z., and A.R. prepared protein samples and performed biochemical experiments. A.R., L.C., L.A.K, H.L and G.L. supervised experiments; H.L. and G.L. conceived the study. All authors contributed to the writing of the manuscript.

## Competing interests

The authors declare no competing interests.
