## [Peer Review File · Nature Communications]

REVIEWER COMMENTS

Reviewer #1 (Remarks to the Author):

In this manuscript "Structures revealing mechanisms of resistance and collateral sensitivity of *Plasmodium falciparum* to proteasome inhibitors", Hsu, Li, Lin, and colleagues present high-resolution cryo-EM structures of Pf20S in complex with TDI-8304. TDI-8304 is a macrocyclic peptide inhibitor recently developed by the authors which selectively targets the *P. falciparum* 20S proteasome and does not target the human 20S proteasome. The structure reveals that TDI-8304 binds both in the β 2 and the β 5 subunit. The authors also show that TDI-8304 kills *P. falciparum* as quickly as chloroquine and artemisinin, two antimalarial drugs that are employed in the clinic, and that it is active against *P. cynomolgi* at the liver stage. The authors also investigate the consequences of a β 6A117D mutation which confers resistance to TDI-8304. This mutation enhance both enzyme inhibition and anti-parasite activity of a tripeptide vinyl sulfone inhibitor WLW-vs. The authors therefore prepared a β 6A117D Pf20S and solved its structure in complex with WLW-vs to elucidate the basis for resistance to TDI-8304 and its enhanced susceptibility to WLW-vs.

Overall the manuscript is well written and all the conclusions drawn by the authors appear conclusive. I will not judge the *in vivo* *P. falciparum* studies which lie outside of my expertise, but will focus myself entirely on the structural biology. Thankfully the authors provided me with the model and the maps of the two complex structures to be able judge them better. Having had a closer look, I did find several inconsistencies which appear to require a major revision of the manuscript:

- First off what made me suspicious are the preparations shown in Supplementary Figure 1, where the authors speak of highly enriched preparations. That statement is justified for the wt Pf20S, but appears quite a stretch for the β 6A117D mutant. The latter preparation is heavily contaminated, leading me to question how valid the structural findings are.
- Second, TDI-8304 structure has an overall resolution of 2.18 by the FSC and a local resolution spread from 2.18 - 3.38 Å. This is also reflected in the locres map in Supplementary Figure 2C. On the other hand the β 6A117D Pf20S structure has a nominal resolution of 2.58 Å and a local resolution spread from 1.79 - 7.22 Å. The locres map in Supplemental Figure 7 of the latter does not reflect this large spread and in fact contradicts this. It seems to me that the quality of the β 6A117D structure is poorer and there are some imprints of over-fitting.
- Third 2.18 Å and 2.6 Å are quite substantial differences in resolution, as 2.2 Å resolution represents a threshold where many aspects of the structure including localised solvent become reliably visible. However, the two maps are indistinguishable in appearance which simply cannot be.
- Fourth, related to my third point, the most critical difference amongst the two structures should reside in the localised solvent. The authors show in Supplemental Table1 that the wt structure has 24 water molecules, whereas β 6A117D has 21. Should these structures really have the resolutions claimed by the authors, one would expect in the order 600 - 1000 water molecules. Moreover, several localised ions should be visible. Both of these are critical for elucidating the structural basis of inhibition and the authors have really foregone an opportunity here. In fact, upon my own visual inspection there are several unmodelled densities in both structures which would represent water molecules and ions. The

conclusion I unfortunately have to reach from this observation is that the authors have done an unsatisfactory job in model building and refinement .

- Fifth, regarding the differences between wt and β 6A117D Pf20S and the structural basis of resistance to TDI-8304: there appears to be a frame-shift in the β 6A117D Pf20S structure. In particular, the stretch from β 6143 - β 6153 seems to be shifted by 12 amino acid which in my view is not justified by the density. Therefore, I have to unfortunately conclude that the interpretation reached by the authors is at least not as clear as presented in the manuscript.

- Sixth, even though the β 6A117D structure is poorer in resolution, and the model resolution is 0.4 Å worse, the model B-factors are lower in the β 6A117D model. That is physically impossible and again points to unsatisfactory model refinement. Clearly also such values are only reached by heavy constraints applied and not by the maps. This is reflected by the bond and angle rms cited by the authors.

In summary, with all the mentioned inconsistencies, I cannot be favourable towards publication of this manuscript in the present form.

Reviewer #2 (Remarks to the Author):

This brief, but interesting and well-written manuscript reports the structure of the Plasmodium falciparum 20S proteasome core particle in complex with two candidate anti-malarial drugs, the macrocyclic peptide TDI-8304 and WLW-vs. The structure of the Pf20S/TDI-8304 complex shows, in great detail, the interactions between TDI-8304 and the β 2 and β 5 subunits, and also reveals an unexpected binding site in the proteasome antechamber. The Pf20S/WLW-vs structure has been reported previously (indeed, it was the first Pf20S structure to be reported), however the structure reported here is at substantially higher resolution and reports the complex with a mutant resistant to TDI-8304 (β 6-A117D). Indeed, the specific structural consequences of the A117D mutation are clear in the two maps, and the Pf20S/WLW-vs structure reveals the mechanism of resistance to TDI-8304 induced by the β 6-A117D mutation. The significant improvement in resolution achieved for both the Pf20S/TDI-8304 and Pf20S/WLW-vs maps will aid the development of novel Pf20S inhibitors by other groups, as atomic-level interrogation of the interactions is made possible by the new maps.

On the whole, I have no major concerns about this manuscript, and think that it will be suitable for publication in Nature Communications, once the below concerns have been addressed.

Major concerns:

- The most intriguing finding of this manuscript is that TDI-8304 binds at an additional site in the core particle antechamber. The authors speculate that unique poly-Asn repeats and a unique surface formed by the Pf subunit. However, these speculations are not supported by experimental data. Did the authors consider attempting to solve a human (Hs) 20S/TDI-8304 structure, to see if it still bound in the antechamber in Hs20S? Such a structure would also potentially provide experimental support for the author's docking studies into the binding mode for TDI-8304 in the Hs20S β 5 active site (Figure 3). While not essential, the ready (commercial) availability of Hs20S would make such a structure determination relatively straightforward and would significantly improve the manuscript.

- I do have some concerns about the sample purity used for the determination of the β 6-A117D structure – the gel does show that the sample was much less pure compared to the WT Pf20S. The sample did reach high resolution regardless. Did the authors perform mass spectrometry to confirm that the proteasome preparations were free of contaminating (human) 20S proteasome?

Minor concerns:

- The Methods note that PDB 6MUW was used as the initial model for building the Pf20S atomic models, however the paper which reported this structure (Xie et al (2019) Nat Microbiol 4: 1990-2000, PMID: 31384003) was not cited. Please cite the manuscript.
- No information is given in the methods about the culturing conditions used to prepare the Pf20S cell pellets for purification.
- Please include map-model FSC curves in Supplementary Figure 2 and Supplementary Figure 7.

Reviewer #3 (Remarks to the Author):

In this manuscript authors disclose cryo-EM structures of the proteasome of the human malaria parasite *Plasmodium falciparum* (Pf20S) with the macrocyclic β 5 inhibitor TDI-8304 and of the mutant Pf20S β 6A117D (resulting from a mutation in the Pf20S β 6 subunit) with the tripeptide vinyl sulfone β 2 inhibitor, WLW-vs. The work was motivated by previous findings suggesting that mutations that confer resistance to β 5 inhibitors result in strains displaying collateral sensitivity by being more susceptible to a β 2 inhibitor, and that co-inhibition of the β 5 and β 2 is synergistic. Authors also present data showing the fast *P. falciparum* killing kinetics of TDI-8304 comparable to chloroquine and artemisinin and activity against *P. cynomolgi* at the liver stage.

The work reported provides an important advance on two levels. First, the disclosed cryo-EM structures shed light on TDI-8304' potency, specificity, and selectivity, as well as the mutant parasite's mechanism of resistance and collateral sensitivity. Second, the cryo-EM structures will find utility in future structure-guided inhibitor design against an important novel antimalarial drug target, Pf20S.

However, one shortcoming of the manuscript is the absence of data from the testing of TDI-8304 against sexual stage gametocyte parasites, which may shed light on the transmission blocking potential of the compound. This is important because one of the stated advantages of targeting Pf20S is activity of inhibitors at multiple stages of the parasite life cycle, including the asexual blood, liver, and sexual gametocyte stage parasites. It is not clear why TDI-8304 was only tested for activity against *P. cynomolgi* at the liver stage. The original publication (Angew Chem Int Ed Engl. 2021 April 19; 60(17): 9279–9283] also only contains asexual blood stage activity data. It is important to test against gametocytes as the data would likely shed light on any differential activity against different life cycle stages of *Plasmodium falciparum* parasites, which in turn might suggest differential target expression levels and/or compound permeability. This is in addition to assessing the potential to contribute to malaria elimination by interrupting transmission.

Point-by-point **responses in blue** to reviewers' comments in black.

Reviewer #1 (Remarks to the Author):

In this manuscript "Structures revealing mechanisms of resistance and collateral sensitivity of Plasmodium falciparum to proteasome inhibitors", Hsu, Li, Lin, and colleagues present high-resolution cryo-EM structures of Pf20S in complex with TDI-8304. TDI-8304 is a macrocyclic peptide inhibitor recently developed by the authors which selectively targets the P. falciparum 20S proteasome and does not target the human 20S proteasome. The structure reveals that TDI-8304 binds both in the $\beta 2$ and the $\beta 5$ subunit. The authors also show that TDI-8304 kills P. falciparum as quickly as chloroquine and artemisinin, two antimalarial drugs that are employed in the clinic, and that it is active against P. cynomolgi at the liver stage. The authors also investigate the consequences of a $\beta 6A117D$ mutation which confers resistance to TDI-8304. This mutation enhances both enzyme inhibition and anti-parasite activity of a tripeptide vinyl sulfone inhibitor WLW-vs. The authors therefore prepared a $\beta 6A117D$ Pf20S and solved its structure in complex with WLW-vs to elucidate the basis for resistance to TDI-8304 and its enhanced susceptibility to WLW-vs.

Overall the manuscript is well written and all the conclusions drawn by the authors appear conclusive. I will not judge the in vivo P. falciparum studies which lie outside of my expertise, but will focus myself entirely on the structural biology. Thankfully the authors provided me with the model and the maps of the two complex structures to be able judge them better. Having had a closer look, I did find several inconsistencies which appear to require a major revision of the manuscript:

We very much appreciate the reviewer's thoughtful comments.

- First off what made me suspicious are the preparations shown in Supplementary Figure 1, where the authors speak of highly enriched preparations. That statement is justified for the wt Pf20S, but appears quite a stretch for the $\beta 6A117D$ mutant. The latter preparation is heavily contaminated, leading me to question how valid the structural findings are.

We agree with the reviewer's comment on the purity of the Pf20S $\beta 6A117D$ and have rephrased as "Pf20S wild type was highly enriched, and Pf20S $\beta 6A117D$ was partially enriched." For the structures of Pf20S and the A117D mutant, the purity of the Pf20S preps was less of a problem for cryo-EM method due to its unique shape. Several published results have demonstrated structural determination using cell-free lysates with limited enrichment (Verbeke, E. J., et al. *Classification of Single Particles from Human Cell Extract Reveals Distinct Structures. Cell Rep* 24, 259-268.e253, 2018; Ho, C. M. et al. *Bottom-up structural proteomics: cryo-EM of protein complexes enriched from the cellular milieu. Nat Methods* 17, 79-85, doi:10.1038/s41592-019-0637-y, 2020). Proteasome particles are of characteristic shape and highly distinguishable. We can easily exclude contaminating proteins from the 20S particles at particle picking and 2D classification stages. The only concern is a potential contamination by human 20S proteasomes in the red blood cells. We minimized this contamination at the stage of cell lysis. After removing red blood cells, the parasite cell pellets were examined carefully, and any tubes showing any hint of red were discarded. Furthermore, minor contaminating 20S particles would be excluded at the heterogeneous classification stage during image process and 3D reconstruction. The ultimate quality control is the EM map features. The human and Pf 20S structures are significantly different at the amino acid level. Our maps fitted only the Pf20S sequences.

However, to fully address the reviewer's concern, we have determined the level of human 20S contamination in our Ps20S preparation. The human c-20S contamination levels in Pf20S WT and

Pf20S β 6A117D are estimated to be at 2.7% and 3.2%, respectively. This data is now included in **Supplementary Fig. 1b**.

- Second, TDI-8304 structure has an overall resolution of 2.18 by the FSC and a local resolution spread from 2.18 - 3.38 Å. This is also reflected in the locres map in Supplementary Figure 2C. On the other hand, the β 6A117D Pf20S structure has a nominal resolution of 2.58 Å and a local resolution spread from 1.79 - 7.22 Å. The locres map in Supplemental Figure 7 of the latter does not reflect this large spread and in fact contradicts this. It seems to me that the quality of the β 6A117D structure is poorer and there are some imprints of over-fitting.

The issues noted by the reviewer are likely due to our use of different programs. The EM map of the wild type Pf20S-TDI-8304 was refined in Relion, and the map of β 6A117D mutant Pf20S was done in cryoSPARC. We used a user-generated mask in Relion but an automatically generated mask in cryoSPARC. As the calculation of local resolution map depends on the half maps and the mask, the generated local resolution ranges were different due to the use of Relion and cryoSPARC for different maps. Both software packages are well accepted and widely used in the structural biology community. To address the reviewer's concern, we have used the atom attributes from the model to determine the valid resolution range for the model. As one can see in the PDB validation reports, the map-to-model correlations at most sequence regions are very high, indicative a high quality and high-resolution structure. We have modified the local resolution figure of β 6A117D to show the full resolution range of the map (**Supplementary Fig. 7**).

- Third 2.18 Å and 2.6 Å are quite substantial differences in resolution, as 2.2 Å resolution represents a threshold where many aspects of the structure including localised solvent become reliably visible. However, the two maps are indistinguishable in appearance which simply cannot be.

We agree the two maps are not significantly different visually. But the resolutions of both maps were calculated similarly by using the gold standard FSC at the 0.143 correlation threshold. The higher resolution map did contain more information that we could appreciate during model building. For example, we could fit the rotamers more accurately in the 2.18 Å map.

- Fourth, related to my third point, the most critical difference amongst the two structures should reside in the localised solvent. The authors show in Supplemental Table1 that the wt structure has 24 water molecules, whereas β 6A117D has 21. Should these structures really have the resolutions claimed by the authors, one would expect in the order 600 - 1000 water molecules. Moreover, several localised ions should be visible. Both of these are critical for elucidating the structural basis of inhibition and the authors have really foregone an opportunity here. In fact, upon my own visual inspection there are several unmodelled densities in both structures which would represent water molecules and ions. The conclusion I unfortunately have to reach from this observation is that the authors have done an unsatisfactory job in model building and refinement.

We have added water molecules in our revised structures based on the size, shape, and contact distance (2.5-3.5 Å) of the densities (218 waters in Pf20S-TDI-8304 structure, 1121 waters in Hs20S-TDI-8304 structure, and 600 waters in the Pf20S β 6A117D-WLWvs structure; see **Supplemental Table 1**). There is little information about ions in the 20S proteasome. To avoid misleading readers, we didn't add ions in our revised structural models.

- Fifth, regarding the differences between wt and β 6A117D Pf20S and the structural basis of resistance to TDI-8304: there appears to be a frame-shift in the β 6A117D Pf20S structure. In particular, the stretch from β 6143 - β 6153 seems to be shifted by 12 amino acid which in my view is not justified by the

density. Therefore, I have to unfortunately conclude that the interpretation reached by the authors is at least not as clear as presented in the manuscript.

We have carefully re-examined the map but could not find a 12-amino-acids frameshift between β 6-residues 143-153 as pointed out by the reviewer. We note that the EM density of Tyr150-Asn151-Tyr152 is of exceptional quality and could have been used as signature to identify the sequence in this region. Furthermore, a 12-amino-acid shift will produce a big unmodeled blob that we do not see in our EM map.

- Sixth, even though the β 6A117D structure is poorer in resolution, and the model resolution is 0.4 Å worse, the model B-factors are lower in the β 6A117D model. That is physically impossible and again points to unsatisfactory model refinement. Clearly also such values are only reached by heavy constraints applied and not by the maps. This is reflected by the bond and angle rms cited by the authors.

As we have mentioned above, the two maps were generated from two different programs (RELION vs cryoSPARC), which likely explains the noted differences due to the different masks and algorithm used in the two programs for resolution estimation. When we re-refined the wild-type Pf20S-TDI-8304 dataset in cryoSPARC, we obtained the EM map at 2.36 Å resolution. And the B-factor was 37.9 Å², which is similar to that of the β 6 A117D structure that was also refined in cryoSPARC.

In the revised manuscript, we have added water molecules to the Pf20S-TDI-8304 model and re-refined the model against the EM map with a mask generated from RELION. The B-factor is now improved to 21.4 Å², which is clearly better than the 43.9 Å² of the lower resolution β 6A117D structure. The use of the program also made the B-factor of Pf20S-TDI-8304 slightly better (21.44 Å²) than that of the high-resolution structure of the Hs20S-TDI-8304 complex (26.1 Å²).

Reviewer #2 (Remarks to the Author):

This brief, but interesting and well-written manuscript reports the structure of the Plasmodium falciparum 20S proteasome core particle in complex with two candidate anti-malarial drugs, the macrocyclic peptide TDI-8304 and WLW-vs. The structure of the Pf20S/TDI-8304 complex shows, in great detail, the interactions between TDI-8304 and the β 2 and β 5 subunits, and also reveals an unexpected binding site in the proteasome antechamber. The Pf20S/WLW-vs structure has been reported previously (indeed, it was the first Pf20S structure to be reported), however the structure reported here is at substantially higher resolution and reports the complex with a mutant resistant to TDI-8304 (β 6-A117D). Indeed, the specific structural consequences of the A117D mutation are clear in the two maps, and the Pf20S/WLW-vs structure reveals the mechanism of resistance to TDI-8304 induced by the β 6-A117D mutation. The significant improvement in resolution achieved for both the Pf20S/TDI-8304 and Pf20S/WLW-vs maps will aid the development of novel Pf20S inhibitors by other groups, as atomic-level interrogation of the interactions is made possible by the new maps.

On the whole, I have no major concerns about this manuscript, and think that it will be suitable for publication in Nature Communications, once the below concerns have been addressed.

We very much appreciate the reviewer's thoughtful comments.

Major concerns:

- The most intriguing finding of this manuscript is that TDI-8304 binds at an additional site in the core particle antechamber. The authors speculate that unique poly-Asn repeats and a unique surface formed

by the Pf subunit. However, these speculations are not supported by experimental data. Did the authors consider attempting to solve a human (Hs) 20S/TDI-8304 structure, to see if it still bound in the antechamber in Hs20S? Such a structure would also potentially provide experimental support for the author's docking studies into the binding mode for TDI-8304 in the Hs20S β 5 active site (Figure 3). While not essential, the ready (commercial) availability of Hs20S would make such a structure determination relatively straightforward and would significantly improve the manuscript.

We agree with the reviewer that finding TDI-8304 in the antechamber of 20S core is interesting. But the biological function of the binding is not important as the activity of TDI-8304 is clearly β 5 inhibition driven with the evidence of β 6A117D mutation conferring marked shift in anti-Pf activity, one of two topics of this manuscript. To protect the integrity of the complete observation, we report the observation. We removed the following line from the manuscript to avoid a potential confusion "It is unclear if these surface loops contribute to the unexpected TDI-8304 binding in the antechamber",

Given the synergy and collateral sensitivity of β 5 inhibition and β 2 inhibition against Plasmodium parasites and cancer cells (relevant, although not related to the scope of this manuscript), we are pursuing β 5 and β 2 co-inhibitors to suppress resistance.

To address the reviewer's concern, we have now determined the structure of human c-20S-TDI-8304 at 2.04 Å resolution. The structure confirms the importance of the pyrrolidinone for the TDI-8304 affinity. Thanks for the good suggestion.

- I do have some concerns about the sample purity used for the determination of the β 6-A117D structure – the gel does show that the sample was much less pure compared to the WT Pf20S. The sample did reach high resolution regardless. Did the authors perform mass spectrometry to confirm that the proteasome preparations were free of contaminating (human) 20S proteasome?

As stated for the same comment from reviewer #1, we rephrased "Pf20S wild type was highly enriched, and Pf20S β 6A117D was partially enriched". With characteristic barrel-shaped, proteasome particles are readily discernible from other contaminating proteins. To further address the reviewer's concern, we have determined the human proteasome contaminations were about 3% in both wild type and mutant preparations. This low level of contamination is negligible in cryo-EM based structure determination.

Minor concerns:

- The Methods note that PDB 6MUW was used as the initial model for building the Pf20S atomic models, however the paper which reported this structure (Xie et al (2019) Nat Microbiol 4: 1990-2000, PMID: 31384003) was not cited. Please cite the manuscript.

Thanks for the reminder. The reference has been added.

- No information is given in the methods about the culturing conditions used to prepare the Pf20S cell pellets for purification.

We have added the reference (Kirkman PNAS).

- Please include map-model FSC curves in Supplementary Figure 2 and Supplementary Figure 7.

We have added in the revised Supplemental figures for the three described structures the map-model FSC curves calculated by the Mtriage in the Phenix package.

Reviewer #3 (Remarks to the Author):

In this manuscript authors disclose cryo-EM structures of the proteasome of the human malaria parasite *Plasmodium falciparum* (Pf20S) with the macrocyclic β 5 inhibitor TDI-8304 and of the mutant Pf20S β 6A117D (resulting from a mutation in the Pf20S β 6 subunit) with the tripeptide vinyl sulfone β 2 inhibitor, WLW-vs. The work was motivated by previous findings suggesting that mutations that confer resistance to β 5 inhibitors result in strains displaying collateral sensitivity by being more susceptible to a β 2 inhibitor, and that co-inhibition of the β 5 and β 2 is synergistic. Authors also present data showing the fast *P. falciparum* killing kinetics of TDI-8304 comparable to chloroquine and artemisinin and activity against *P. cynomolgi* at the liver stage.

The work reported provides an important advance on two levels. First, the disclosed cryo-EM structures shed light on TDI-8304' potency, specificity, and selectivity, as well as the mutant parasite's mechanism of resistance and collateral sensitivity. Second, the cryo-EM structures will find utility in future structure-guided inhibitor design against an important novel antimalarial drug target, Pf20S.

However, one shortcoming of the manuscript is the absence of data from the testing of TDI-8304 against sexual stage gametocyte parasites, which may shed light on the transmission blocking potential of the compound. This is important because one of the stated advantages of targeting Pf20S is activity of inhibitors at multiple stages of the parasite life cycle, including the asexual blood, liver, and sexual gametocyte stage parasites. It is not clear why TDI-8304 was only tested for activity against *P. cynomolgi* at the liver stage. The original publication (Angew Chem Int Ed Engl. 2021 April 19; 60(17): 9279–9283] also only contains asexual blood stage activity data. It is important to test against gametocytes as the data would likely shed light on any differential activity against different life cycle stages of *Plasmodium falciparum* parasites, which in turn might suggest differential target expression levels and/or compound permeability. This is in addition to assessing the potential to contribute to malaria elimination by interrupting transmission.

We very much appreciate the reviewer's thoughtful comments. However, we and others have established that proteasome inhibitors are active against gametocytes at all stages, and active against male gamete activation, but not female gamete activation (Kirkman PNAS).

REVIEWERS' COMMENTS

Reviewer #1 (Remarks to the Author):

In this revised version of the manuscript, the authors have addressed most of my concerns. Also, the addition of the c20S proteasome structure has clarified some issues and is appreciated. As a minor comment: please avoid stating resolutions with two decimal points. It gives the appearance of accuracy and precision which the structure does not have at $\sim 2 \text{ \AA}$. Rather it is advisable to round up to the next decimal point.

With this I congratulate the authors on an interesting and accomplished manuscript!

Reviewer #2 (Remarks to the Author):

I thank the authors for their thoughtful response to my comments, in particular I acknowledge the experimental work involved in solving the Hs20S/TDI-8304 structure. My major comments from the previous round of review have been adequately addressed. Several minor comments (largely small formatting suggestions and minor typographical errors) are noted below. In particular, the authors should pay attention to correct use of definite (the) and indefinite (a/an) articles.

The article does not need another round of revision, and will be suitable for publication once the below comments have been addressed.

Minor comments:

- * Section 2: 'reported in a medium resolution range', should be 'reported at medium resolution'.
- * Section 2: 'A 20S proteasome core particle', 'The 20S proteasome core particle'.
- * Section 2: 'TDI-8304 binds to the $\beta 5$ with the...' should be 'TDI-8304 binds to the $\beta 5$ subunit with the P1 cyclopentyl'.
- * Section 2: 'contrast to the potent inhibition of $\beta 5$ ' should be 'contrast to the potent inhibition of the $\beta 5$ chymotrypsin-like activity'.
- * Section 3: 'selective mechanism against human proteasome' should be 'selective mechanism against the human proteasome'.
- * Discussion: 'TDI-8304 also binds to $\beta 2$ active subunit' should be 'TDI-8304 also binds to the $\beta 2$ active subunit'.
- * I would suggest referring to the constitutive human 20S proteasome core particle as 'cHs20S' in the manuscript - I feel that 'c20S' is a little unclear and could be made more specific (for example, in Figure 3C-E and in section 3).
- * EMDB deposition numbers must be noted in the 'Data Availability' section.
- * Supplementary Figure 1 caption - 'Image J' should be 'ImageJ'.

* If the micrographs in Supplementary Figures 2, 5 and 9 are low-pass filtered, this should be noted (particularly Supplementary Figure 9A, as the contrast between the proteasome particles and background seems much better compared to the others).

Re: NCOMMS-23-23737B

Response to reviewers' further comments.

We thank both reviewers' support and careful reading and suggestions, which made this manuscript a stronger paper.

Reviewer #1 (Remarks to the Author):

In this revised version of the manuscript, the authors have addressed most of my concerns. Also, the addition of the c20S proteasome structure has clarified some issues and is appreciated. As a minor comment: please avoid stating resolutions with two decimal points. It gives the appearance of accuracy and precision which the structure does not have at $\sim 2 \text{ \AA}$. Rather it is advisable to round up to the next decimal point.

Thanks for the suggestions, but we think we should keep the two decimals as our structures are of relatively high resolution. Also, several proteasome structures at this resolution range were reported with two decimals, for example: 5LF3 (human constitutive proteasome with bortezomib) 2.10 \AA ; 7AWE (human immunoproteasome with M3258) 2.29 \AA ; 5LEZ (human constitutive proteasome with bortezomib Oprozomib) 2.19 \AA .

With this I congratulate the authors on an interesting and accomplished manuscript!

Greatly appreciate it.

Reviewer #2 (Remarks to the Author):

I thank the authors for their thoughtful response to my comments, in particular I acknowledge the experimental work involved in solving the Hs20S/TDI-8304 structure. My major comments from the previous round of review have been adequately addressed. Several minor comments (largely small formatting suggestions and minor typographical errors) are noted below. In particular, the authors should pay attention to correct use of definite (the) and indefinite (a/an) articles.

The article does not need another round of revision, and will be suitable for publication once the below comments have been addressed.

Thank you for the support and suggestions of fine English grammar. Appreciate it greatly.

Minor comments:

* Section 2: 'reported in a medium resolution range', should be 'reported at medium resolution'.

Fixed.

* Section 2: 'A 20S proteasome core particle', 'The 20S proteasome core particle'.

Fixed.

* Section 2: 'TDI-8304 binds to the $\beta 5$ with the...' should be 'TDI-8304 binds to the $\beta 5$ subunit with the P1 cyclopentyl'.

Fixed.

* Section 2: 'contrast to the potent inhibition of $\beta 5$ ' should be 'contrast to the potent inhibition of the $\beta 5$ chymotrypsin-like activity'.

Fixed.

* Section 3: 'selective mechanism against human proteasome' should be 'selective mechanism against the human proteasome'.

Fixed.

* Discussion: 'TDI-8304 also binds to $\beta 2$ active subunit' should be 'TDI-8304 also binds to the $\beta 2$ active subunit'.

Fixed.

* I would suggest referring to the constitutive human 20S proteasome core particle as 'cHs20S' in the manuscript - I feel that 'c20S' is a little unclear and could be made more specific (for example, in Figure 3C-E and in section 3).

Thanks for the suggestion. We have been using c-20S for human constitutive proteasome in several papers, and we used the full name of the human constitutive proteasome with the abbreviation (c-20S) in the text when we used it the first time.

* EMDB deposition numbers must be noted in the 'Data Availability' section.

The EMD codes were added in the 'Data Availability' section and Supplemental Table 1.

* Supplementary Figure 1 caption - 'Image J' should be 'ImageJ'.

Fixed.

* If the micrographs in Supplementary Figures 2, 5 and 9 are low-pass filtered, this should be noted (particularly Supplementary Figure 9A, as the contrast between the proteasome particles and background seems much better compared to the others).

We didn't use low-pass filter for the Supplementary Figure 9A. All the raw images were selected from Cryosparc after importing the micrographs. In fact, we found the contrast of c-20S-TDI-8304 grid was especially good while performing test collection. Thus, we changed the defocus range from -1.3 - -1.8 to -1.0 - -1.4.